# PROGRESSIVE UPSAMPLING AUDIO SYNTHESIS VIA EFFICIENT ADVERSARIAL TRAINING

## ABSTRACT

This paper proposes a novel generative model called PUGAN, which progressively synthesizes high-quality audio in a raw waveform. PUGAN leverages on the recently proposed idea of progressive generation of higher-resolution images by stacking multiple encode-decoder architectures. To effectively apply it to raw audio generation, we propose two novel modules: (1) a neural upsampling layer and (2) a sinc convolutional layer. Compared to the existing state-of-the-art model called WaveGAN, which uses a single decoder architecture, our model generates audio signals and converts them in a higher resolution in a progressive manner, while using a significantly smaller number of parameters, e.g., 20x smaller for 44.1 kHz output, than an existing technique called WaveGAN. Our experiments show that the audio signals can be generated in real-time with the comparable quality to that of WaveGAN with respect to the inception scores and the human evaluation.

## 1 INTRODUCTION

Synthesis of realistic sound is a long-studied research topic, with various real-world applications such as text-to-speech (TTS) (Wang et al., 2017; Ping et al., 2018), sound effect (Raghuvanshi et al., 2016), and music generation (Briot et al., 2017; Dong et al., 2018; Huang et al., 2019). Various techniques have been developed ranging from the sample-based to more computational ones such as the additive/subtractive synthesis, frequency modulation granular synthesis, and even a full physics-based simulation (Cook, 2002). Human's audible frequency range is up to 20 kHz, so the standard sampling rate for music and sound is 44.1 kHz. Thus, for interactive applications and live performances, the generation of the high temporal-resolution audio (i.e., 44.1 kHz) in real-time has to meet the standard of human perceptual sensitivity to sound. However, the aforementioned methods often fail to do so, due to their heavy computational complexity with respect to the data size. Because of this, professional sound synthesizers usually have no choice but to rely on hardware implementations.(Wessel & Wright, 2002)

Generative adversarial networks (GANs) (Goodfellow et al., 2014) have emerged as a promising approach to the versatile (e.g., conditional generation from a low-dimensional latent vector (Mirza & Osindero, 2014)) and high-quality (e.g., super-resolution GAN (Ledig et al., 2017)) image. One of the first GAN models for sound synthesis have been designed to first produce the spectrogram (or some other similar intermediate representations) (Donahue et al., 2019; Engel et al., 2019; Marafioti et al., 2019). A spectrogram is a compact 2D representation of audio signals in terms of its frequency spectrum over time. The spectrogram can then be converted into the estimated time-domain waveform using the Griffin & Lim algorithm (Griffin & Lim, 1984). However, such a conversion process does not only introduces nontrivial errors but also runs slowly, preventing the approach from being applied at an interactive rate[1] . WaveGAN (Donahue et al., 2019) was the first and state-of-the-art GAN model that can generate raw waveform audio from scratch.

The first generations of sound-generating GANs, like the WaveGAN and its followers, have been influenced much by the enormously successful generative models for image synthesis. They can

---

[1]The interactive rate refers to the the maximum temporal threshold of around 10msec (Wessel & Wright, 2002) over which humans would not be able to recognize the sound making event and the resultant sound as occuring at the same time.

be divided into those that employ the single decoder architecture (e.g., DCGAN and StyleGAN (Radford et al., 2016; Karras et al., 2019)) and those that encode and decode the intermediate representations in several and progressive stages (e.g., StackGAN and progressive GAN (Zhang et al., 2017; Karras et al., 2018)). WaveGAN is the direct descendant of DCGAN with modification for the 1D audio data, while GANSynth applied the concept of progressive generation of audio, but using the 2D spectrogram, treating the audio as a 2D image. No previous work in GAN based audio generation has attempted the direct and fast synthesis of 1D raw audio waveform employing the multiple and progressive encoder-decoder architecture.

Therefore, in this paper, we propose PUGAN, modification and extension of WaveGAN architecture for efficiently synthesizing raw-waveform audio through progressive training. PUGAN generates low sampling rate audio using the first few layers of the original WaveGAN (referred to as the lightweight WaveGAN module). The latter layers of WaveGAN are replaced with the bandwidth extension modules, each of which is composed of the neural upsampling layer and encoder/decoder. They progressively output (progressively trained too) the higher sampling rate audio. For the effective progressive training and generation, instead of the usual upsampling method such as the nearest neighbor used in image generation, PUGAN uses a new upsampling methods often employed in the digital signal processing (DSP) field in an attempt to preserve the frequency information of the original data (Oppenheim, 1999). This upsample process consists of the zero insertion and 1D convolution to function as an interpolation infinite impulse response (IIR) filter. On the discriminator side, we add the Sinc convolution (Ravanelli & Bengio, 2018) before the first layer to replicate the function of the parameterized low pass Sinc filter, also a popular technique in the DSP area. We have also evaluated PUGAN in terms of both quantitative computational performance and qualitative metrics including the human perceptual evaluation. (demo and code: https://pugan-iclr-demo.herokuapp.com/)

Overall, our contributions include the following:

- propose PUGAN, with novel neural modules (upsampling and bandwidth extension) for the efficient generation of raw waveform audio,
- apply the concept of resampling (in the generator) and sinc convolution layers (in the discriminator) suitable for handling sound generation instead of the conventional upsampling or convolution methods, and
- demonstrate the effectiveness of the proposed approach by generating raw waveform audio with significantly less number of parameters in real-time with equivalent output quality as WaveGAN.

## 2 RELATED WORK

We first review related research in two areas, namely, the GAN-based sound generation and audio-to-audio conversion.

### 2.1 GAN BASED AUDIO GENERATION

WaveGAN (Donahue et al., 2019) and GANSynth (Engel et al., 2019) are the two recent notable work that have applied the GAN technique to sound effects generation for the first time. WaveGAN modified the DCGAN and took the approach to operate for and generate one dimensional sound data (raw-waveform) fast and directly (and distinguishing itself from the work like the SpecGAN (Donahue et al., 2019) which used the usual 2D/image-based processing and spectrogram output representation). WaveGAN also added the phase shuffle module to prevent the discriminator from learning the checkerboard artifact, and post-processing convolution layer with a relatively wide kernel size for noise reduction.

GANSynth generated sound effects through the spectrogram-like representation, but its output quality was satisfactory for only pure tone instrumental sounds. TiF-GAN (Marafioti et al., 2019) made a marginal improvement by adding a provisional step for the phase information reconstruction. Note that in the generative setting, the 2D based approach (using representations like spectrograms) is considered problematic as spectrograms are not fully invertible to sound without a loss (thus inexact) and the inversion by, say, the most popular Griffin & Lim algorithm is time-consuming.

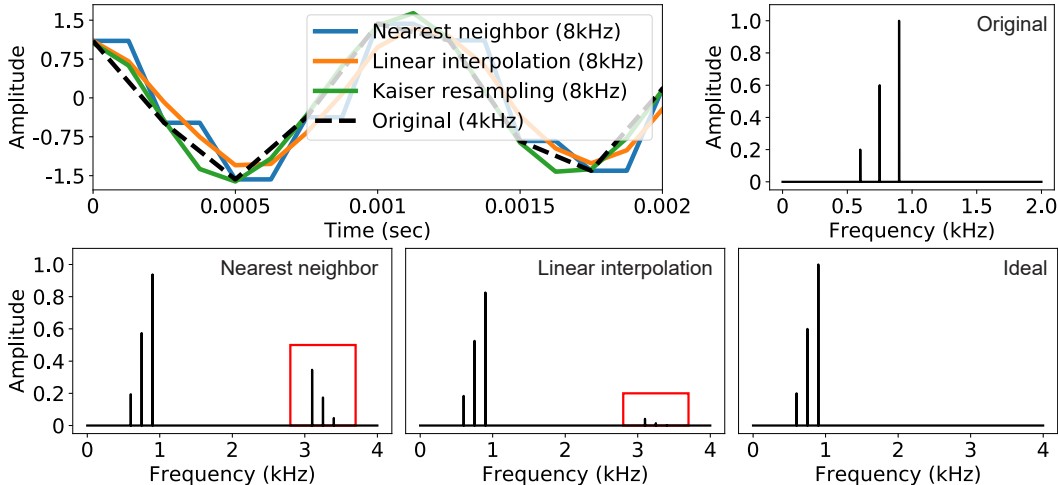

Figure 1: An original signal composed of three frequency components at 600, 750 and 900Hz - time domain plot (top left) and frequency domain (top right) plots. The plots below show the upsampling results using the nearest neighbor, linear interpolation and Kaiser resampling methods. The examples illustrate the occurrences of high frequency sidebands (noise) when the first two simpler upsampling methods are used. This may be problematic if the high resolution output is required. Kaiser resampling is regarded as ideal result (bottom right) and we use neural upsampling layer to minimize noise.

A fast version of Griffin & Lim algorithm (Perraudin et al., 2013) and Deep Griffin & Lim iteration approaches (Masuyama et al., 2019) have been proposed to improve the inversion performance; they have not reached the aforementioned level required for interactive applications. In addition, all previous GAN-based sound synthesizers were configured and experimented to output up to only 16 kHz audio. Such a sampling rate is sufficient for general TTS systems, but, as noted earlier, not for general sound effects or music. WaveNet (Oord et al., 2016) can generate 44.1 kHz sampling rate audio, but aside from the questionable quality of the output, its autoregressive nature make it difficult to achieve real-time performance through the GPU parallelization. In this regard, once trained, the generator part of GAN generally can operate faster. But for WaveGAN to generate 44.1 kHz audio, additional transposed convolutional layers (to the current 16 kHz generator) would make the architectural parameters prohibitively large (two times higher) and likewise the generation time.

## 2.2 AUDIO-TO-AUDIO CONVERSION

Audio-to-audio conversion refers to the task of taking an input audio sample and converting it into another with different characteristics. Most deep learning based audio conversion models have been influenced by similar image-to-image translation research. For instance, CycleGAN-VC (Kaneko & Kameoka, 2018), StarGAN-VC (Kameoka et al., 2018), and WaveCycleGAN (Tanaka et al., 2018) all looked into the problem of voice conversion, and were based on the previous works of CycleGAN (Zhu et al., 2017) and StarGAN (Choi et al., 2018). The task of denoising (Pascual et al., 2017) or generation of super-resolution signal (Eskimez & Koishida, 2019) can also be regarded as a form of signal (or audio) conversion. Recently, few attempts have been made to apply GAN to the task of bandwidth extensions such as the SSRGAN (Eskimez & Koishida, 2019; Li et al., 2019). Note that our objective is the generation of high-resolution audio and sound effects rather than just conversion.

## 3 DATA CHARACTERISTICS: AUDIO VERSUS IMAGE

In this section, we discuss potential reasons of why conventional GAN architectures have been successful in generating 2D images (Zhu et al., 2017; Karras et al., 2018; 2019), but less so for 1D sound waves with respect to their data characteristics. This analysis can give us hints on how to newly configure the GAN architecture to generate the sound signal faster with higher quality.

Image and sound, both as signals, contain information across the frequency domain. Sound has the added dimension of time. Humans are highly sensitive to variation over time of the sound content over all the frequency range, which makes its quality depend on reproduction of all frequency components. In other words, in sound, the different frequency range may represent a particular characteristic (e.g., low bass male sound vs. high pitch female sound) (Klevans & Rodman, 1997). In contrast, in images, high resolution components often correspond to details or even noises, and as such static image recognition and understanding may depend less on them(Heittola et al., 2009).

Upsampling of the data are important parts of the GAN architecture (especially with respect to the conversion process). In image generation, for the reason mentioned above, the upsampling by standard interpolation, such as the nearest neighbor or linear interpolation, may suffice. Fig. 1 compares the application of the simple nearest neighbor based upsampling and linear interpolation to the sinc function based upsampling (or resampling as better known in the DSP area). Fig. 1 shows the upsampling results using the nearest neighbor, linear interpolation and Kaiser resampling methods. The examples illustrate the occurrences of high frequency sidebands (noise) when the first two simpler upsampling methods are used. This may be particularly problematic if the high-resolution output is required.

Another possibly effective method for dealing with signals of multi-frequency components is the resolution-wise progressive generation (and training) technique, as was demonstrated by the work of Progressive GAN (Karras et al., 2018). While the original Progressive GAN was applied for 2D images, and similarly to spectrogram generation, we have applied the same idea to the 1D audio signal. However, the preliminary pilot result was not satisfactory; the reconstructed results were unnaturally smooth in the high frequency range. This is attributed to the similar reason, the stride-1 transposed convolution layer effectively acting as a simple moving averaging method.

On the other hand, for an audio generation as WaveGAN has implemented, the upsampling based on the transposed convolution is more proper than the others such as nearest neighbor. It was deemed more accurate in "capturing" (filtering out) the frequency-wise characteristics in the generation process in comparison to using the nearest neighbor. The only problem may be the fact that the number of the relevant architectural parameters grows excessively according to the output size, which ultimately would render the generation process non real-time.

To summarize, based on these observations, the newly proposed PUGAN architecture would proceed to first train to learn the gross structure of the aural information distribution and fast produce the low resolution audio, then incrementally convert and enrich the output to a higher resolution efficiently instead of having to deal with the entire scale space with computationally heavy architecture simultaneously.

## 4 PUGAN: PROGRESSIVE UPSAMPLING GAN

In this section, we explain the details of the PUGAN, as also shown in Fig. 2. Note that the objective of the proposed design is to produce 44.1 kHz raw audio waveform with reasonable quality in real-time deployable for interactive and live applications.

The Generative Adversarial Network (GAN) can be applied to generate probablistic solutions to a domain problem by framing it as a supervised learning problem with two sub-models: the generator model that one trains to generate new examples, and the discriminator model that tries to classify examples as either real (from the domain) or fake (generated). The two models are trained together in a zero-sum game, adversarial, until the discriminator model is fooled about half the time, meaning the generator model is generating plausible examples. The details of the GAN architecture is omitted and referred to (Goodfellow et al., 2014).

### 4.1 PROBABILISTIC MODEL

The probablistic model of our raw waveform generation can be stated as below. We denote the set of audio data downsampled from the maximum sampling rate of $B_0$, $\mathbf{x}^{B_0}$, successively $\mathbf{n}$ times as $\mathbf{X} = (\mathbf{x}^{B_0}, \mathbf{x}^{B_1}, ..., \mathbf{x}^{B_n})$ and consider the joint probability of $\mathbf{X}$. That is, the audio of resolution $\mathbf{i}$ is dependent on all its lower resolution data, $p(\mathbf{X}) = \prod_{i=0}^{n} p(\mathbf{x}^{B_i}|\mathbf{x}^{B_{i+1}}, ..., \mathbf{x}^{B_n})$, or simplified as being dependent only on its immediate predecessor, $p(\mathbf{X}) = \prod_{i=0}^{n} p(\mathbf{x}^{B_i}|\mathbf{x}^{B_{i+1}})$.

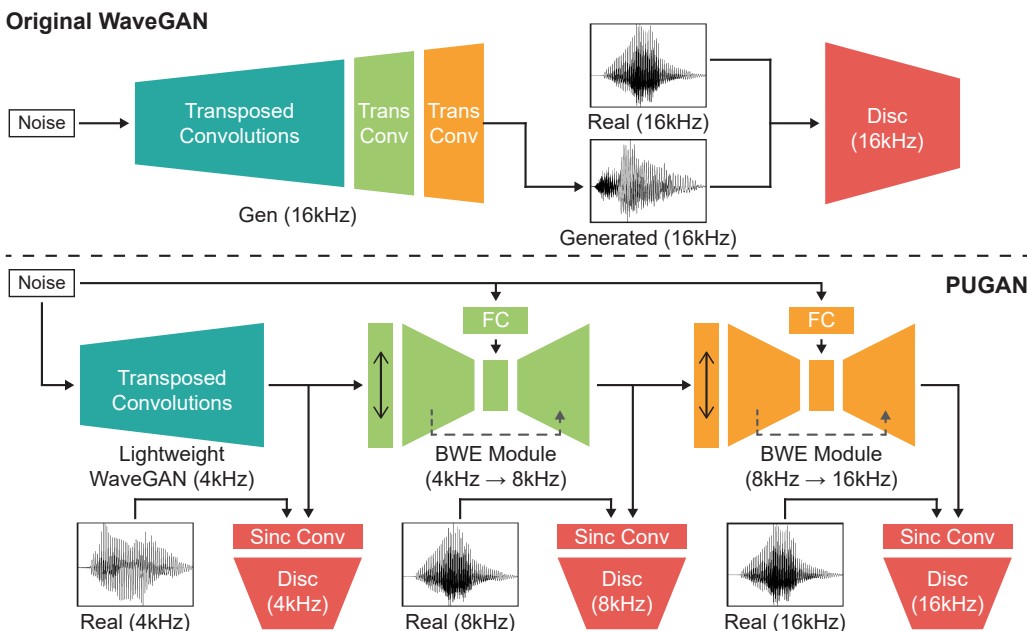

Figure 2: Overview of PUGAN. The generator is composed of the lightweight Wavegan Module for first generating the low sampling rate audio and several encoder-decoder architectures called "band-width extension module (BWE)", which upsample the input to a high sampling rate and enriches the content. We eliminate the number of layers from WaveGAN and instead attach a series of BWEs. In BWE, the neural upsampling layer (indicated with the with up and down bidirectional arrow) is trained to preserve the frequency information of the original data.

## 4.2 GENERATOR

The generator part of PUGAN is composed of the lightweight WaveGAN module for first generating the low resolution waveform, and a series of U-net based "bandwidth extension modules (BWE)" each of which upsample the input to a higher resolution (up to 44.1 kHz) and enriches the content. In other words, the transposed convolutional layers in the original WaveGAN generator are replaced with the BWE modules.

### 4.2.1 LIGHTWEIGHT WAVEGAN MODULE

The original WaveGAN generator synthesizes one second of 16 kHz sampling rate audio. We reduce the number of transposed convolutional layers and decrease the sampling rate of the output audio by a factor of four (4 kHz). For the later experimental purpose, we have implemented other versions that would produce 2 kHz and 8 kHz as well by adjusting the noise input dimension accordingly. The output from the lightweight WaveGAN module becomes the input for the next step, the BWE. Therefore, depending on the output size of this WaveGAN module, the number of subsequent BWE's would differ (e.g., with 4 kHz lightweight WaveGAN module, two BWEs for 16 kHz or 4 BWEs for 44.1 kHz output).

### 4.2.2 BANDWIDTH EXTENSION MODULE (BWE)

The bandwidth extension module plays the role of inserting and adding high frequency information into the input audio. The module has a neural upsampling layer and an encoder-decoder architecture for conversion. In the default configuration, the upsampling unit doubles the sampling rate of the input audio. Moreover, no information loss from the lower resolution data will occur and the upsampling unit is to emulate the ideal windowed sinc filter with its width parameter tuned to the nature of the audio signal. Note that the convolution computation used in the deep learning architectures is mathematically equivalent to y-axis symmetric cross-correlation and also the sinc filter.

| Sampling rate | Architecture | No. of Params. (in mill.) | Ratio | | | | | |
|---|---|---|---|---|---|---|---|---|
| | | | Lightweight WaveGAN | Number of BWE module | | | | |
| | | | | 1 | 2 | 3 | 4 | |
| 16 kHz | WaveGAN | 19.69 (100%) | 100% | - | - | - | - | |
| | PUGAN with one BWE | 2.33 (11.8%) | 92.4% | 7.6% | - | - | - | |
| | PUGAN with two BWEs | 1.96 (9.9%) | 82.8% | 8.2% | 9.0% | - | - | |
| 44.1 kHz | WaveGAN | 46.00 (100%) | 100% | - | - | - | - | |
| | PUGAN with four BWEs | 2.44 (5.3%) | 66.4% | 6.5% | 7.2% | 8.6% | 11.3% | |

Table 1: The number of parameters among the compared architectures, different configurations and by output resolution (WaveGAN, PUGAN with one BWE, PUGAN with two BWEs and PUGAN with four BWEs for two sampling rates, 16 kHz and 44.1 kHz). Note that all varied configurations have significantly less number of parameters than the comparable WaveGAN. Within the PUGAN, also note that most of the parameters are subsumed in the lightweight WaveGAN module. WaveGAN was made to generate the 1.48 second long 44.1 kHz audio as 4-second long 16 kHz audio.

The number of architectural parameters in WaveGAN is approximately geometrically proportional to the product of the input and output channels and the needed transposed convolutional layers. Compared to the 16 kHz WaveGAN, its 44.1 kHz version would increase this complexity 2-fold, while PUGAN with equivalent output would possess only about 5% of this figure respectively. See Table. 1 for a more detailed description and comparison.

The number of parameters of a bandwidth extension module does not change except for the fully connected layer. Thus, the number of parameters in the entire model increases in proportion to the number of modules. Also, the bandwidth extension module itself has fewer parameters compared to equivalent WaveGAN based generator layer. When we measure the ratio of the number of parameters of each module in PUGAN, the bandwidth extension module is highly efficient because the lightweight WaveGAN module accounts for up to 90% of the total.

## 4.3 DISCRIMINATOR

As we focus on improving the generator performance for real-time application, we opt to use the same discriminator architecture of WaveGAN with the aforementioned phase shuffle module. As indicated in our data characteristics observation in Section 3 and demonstrated in the work of SincNet (Ravanelli & Bengio, 2018), we added a sinc convolutional layer in the discriminator to help the module learn and discover more meaningful and effective features. On every layer in the discriminator, we also added the spectral normalization (Miyato et al., 2018) which is a well-known technique to stabilize the discriminator by restricting the Lipschitz constant. To demonstrate the significance of both modules, we created Improved WaveGAN that uses those modules in the discriminator and compared the performance with other models.

There are separate discriminators for each generator module (outputting intermediate and final audio at different sample rates), namely the lightweight WaveGAN module and BWE's.

## 5 EXPERIMENT

We introduce dataset and training process, and demonstrate how to evaluate PUGAN in terms of both quantitative computational performance and qualitative metrics including the human perceptual evaluation.

### 5.1 DATASET

The Speech Commands (Warden, 2018) dataset is a collection of voice command recordings from various speakers. The length of each audio sample is under one second, containing only one word. Similarly to WaveGAN and TiFGAN, we used a subset of the dataset, sounds of ten-digit commands (i.e., "zero", "one", etc.). The training does not involve any phonology information as our evaluation only concerns the generation as sound effects. The sampling rate of the data is 16 kHz, and the number of total training samples is about 18,000. We reproduced the under-sampled data from the

original as needed by the progressive training in PUGAN. As the undersampling method can affect the structure of waveform, we used the *LibROSA* library (McFee et al., 2015) to resample the audio and applied the *kaiser-best* method.

## 5.2 TRAINING

The WGAN-GP and Adam optimizer (Kingma & Ba, 2015) was used as the loss function and optimization algorithm in our model training. As our model is progressive, and likewise the training process; as the output resolution doubles and new generators are learned. Also, the learning of the previous modules continues to update their parameters. However, the discriminators corresponding to each generator stop learning once the resolution level is increased. We will cover the training details in the Appendix.

## 5.3 INCEPTION SCORE

The Inception Score (IS) (Salimans et al., 2016) is a well-known metric for assessing the quality of the data generated from GAN. It utilizes the classification model and computes the KL divergence of classification probabilities between ground truth and generated data. We evaluated PUGAN comparatively, based on IS, to the baseline model (WaveGAN) by employing the pretrained network from the official WaveGAN repository[2].

## 5.4 HUMAN EVALUATION

We created 18,000 audio samples (16 kHz) of real data, PUGAN (with 1 BWE), WaveGAN and improved (by the authors) WaveGAN for comparative human evaluation. The data were labeled (i.e., which digit) by using the pretrained WaveGAN. Three hundred samples per class in the order of the prediction probabilities from the pretrained classifier were chosen to be presented to the 14 human subjects. In the first session, pair-wise comparison tests were conducted for the six combinations; the subject was to choose the one perceived as having better subjective quality. In the second session, 80 audio samples selected in a balanced fashion from the four data categories (ground truth, PUGAN, WaveGAN, improved WaveGAN) were presented to the subjects who were asked to identify the class labels. The task accuracy was recorded along with subjective quality ratings.

## 6 RESULTS AND DISCUSSION

In this section, we discuss the quantitative and qualitative evaluation and the results show the superiority of our proposed model compared to the existing models.

### 6.1 IS AND HUMAN EVALUATION

Table. 2 shows the results of the qualitative and quantitative evaluation. We compared the IS and other subjective quality metrics among the original WaveGAN, an improved WaveGAN (which contains spectral normalization and sinc convolution in the discriminator) and the varied configurations PUGAN. IS was measured using the pretrained WaveGAN model from the official repository. Human evaluation includes the accuracy of identifying the correct label and subjective rating of the sound quality in the scale of 1 to 5.

Compared to the original WaveGAN, the improved one with spectral normalization and sinc convolution exhibited an increase in the IS, and likewise for the accuracy and subjective quality. This implies for the positive effect of the use of sinc convolution, which is also used equally in the lightweight WaveGAN module of PUGAN. We varied the PUGAN in terms of the sampling rate of the intermediate low-resolution data produced by the lightweight WaveGAN and subsequently the number of required BWE modules to finally produce 16 kHz or 44.1 kHz output, namely, BWE-1 and BWE-2. Regardless of the variation, the table shows that PUGAN resulted in the higher IS than WaveGAN. A trend of increasing IS for less number of BWE modules was observed, and this seems to be attributed to the trade-off between the number of architectural parameters and output

---

[2]https://github.com/chrisdonahue/wavegan

| Architecture | Inception score | Accuracy | Quality |
|---|---|---|---|
| Real (Train) | 9.18 ±0.04 | 0.98 | 4.8 ±0.7 |
| Real (Test) | 7.98 ±0.20 | - | - |
| WaveGAN | 2.65 ±0.03 | 0.63 | 2.6 ±1.3 |
| + specnorm | 3.11 ±0.03 | - | - |
| **+ sinc conv** | **3.41** ±0.04 | **0.52** | **2.7** ±1.5 |
| **PUGAN with one BWE** | **4.63** ±0.05 | **0.76** | **3.4** ±1.2 |
| PUGAN with two BWEs | 3.94 ±0.04 | - | - |
| PUGAN with three BWEs | 3.80 ±0.08 | - | - |

Table 2: Result of qualitative and quantitative evaluation. We compared the IS and other subjective quality metrics among the original WaveGAN, an improved WaveGAN (which contains spectral normalization and sinc convolution in the discriminator) and the varied configurations of PUGAN. IS was measured using the pretrained WaveGAN model from the official repository. Human evaluation includes the accuracy of identifying the correct label and subjective rating of the sound quality in the scale of 1 to 5.

| Architecture | wins | vs. Real | vs. WaveGAN | vs. Improved WaveGAN | vs. PUGAN with one BWE |
|---|---|---|---|---|---|
| Real | 793 | - | 95% | 93% | 91% |
| WaveGAN | 210 | 5% | - | 44% | 24% |
| Improved WaveGAN | 274 | 7% | 56% | - | 40% |
| PUGAN with one BWE | 403 | 9% | 76% | 60% | - |

Table 3: Number of wins on the pair-wise comparison among real data, original WaveGAN, the improved WaveGAN (which contains spectral normalization and sinc convolution in the discriminator) and PUGAN resulting highest inception score.

quality. Further research will be needed to find the right balance on the division of labor between the lightweight WaveGAN and the BWE modules in the regard.

The subjective human evaluation was carried out using PUGAN with one BWE (who scored the highest IS), and in all aspects, PUGAN showed better subjective ratings (accuracy and subjective quality). Table. 3 shows the pair-wise comparison. PUGAN also greatly improved the number of wins against the WaveGANs. Against the ground truth data, PUGAN improved the number of wins up to 9% compared to the 5% mark by WaveGAN. However, in absolute scale, it also implies that there remains much room (and future research) for further improvement in the output quality. In the same vein, several participants reported in the post-briefing that certain sounds (e.g., "six") were heard clearer that others, and sounds seemingly juxtaposed with the few (e.g., "four" and "one" mixed up together) were sometimes perceived. These could be artifacts from generating the samples without any conditioning (vs. using the phonetics in the context of usage for TTS as done in the WaveGAN evaluation).

## 6.2 COMPUTATION COST

One of the primary concerns in our work is whether the proposed model can properly synthesize the sound samples in real-time. The maximum temporal threshold is at around 7-10msec upon the sound making the event; under this threshold, it is known humans would recognize the sound-making event and the resultant sound as separate events. We compared the computational cost (time) among the various configurations of our model and WaveGAN. The comparison was made for the synthesis of (1) one second long 16 kHz sampling rate audio samples and (2) 1.48 seconds long 44.1 kHz sampling rate audio samples. The execution time measurements were repeated 100 times and the average was taken. The computing hardware used for the comparison was a workstation with the Intel Core i9-7920X CPU (with 32GB RAM), and using the NVidia TITAN RTX GPU.

For the 16 kHz audio, PUGAN could generate 100 samples (in a batch) in 0.56 sec. using just the CPU and 0.005 sec. when the GPU was used. The WaveGAN took 1.19 sec. and 0.06 sec. respectively. For the 44.1 kHz audio samples, PUGAN took 2.47 sec. and 0.006 sec., while WaveGAN required 4.4 sec. and 0.18 sec. For 44.1 kHz audio, 100 samples in 2.47 sec. or 0.006 sec. translates

to 24.7 msec or 0.06 msec of latency for short sound effects (e.g. "bang", "clunk"), barely sufficient for real time usage. On the other hand, WaveGAN, twice as slower, will not meet the real time requirement for interactive high resolution (44.1 kHz) audio generation.

## 6.3 CONCLUSIONS AND FUTURE WORK

In this paper, we proposed a novel GAN model based on a stacked encoder-decoder architecture for high-quality real-time audio generation from scratch. Inspired by signal processing literature, the key to success lies in our neural upsampling layer and sinc convolution, allowing, for the first time, the progressive growth of high frequency audio signals with a significantly lightweight architecture, compared to existing state-of-the-art methods such as WaveGAN.

As future work, we plan to improve a discriminator architecture in a progressive manner so that we can properly generate realistic audio signals in a longer time span. On the other hand, we will also explore the applicability of our proposed neural upsampling layer to image generation models in computer vision domains.

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

## A  APPENDIX

### A.1  ARCHITECTURE DETAILS

We implemented WaveGAN using PyTorch and modifed to lightweight wavegan generator. In Table **??**, n is the batch size, m and l increase in proportion to sampling rate of output audio and d is the the number of channel. We attached sinc convolution to WaveGAN discriminator with no changes.

| Operation | Kernel Size | Output Shape |
|---|---|---|
| Input $z \sim \mathcal{U}(-1, 1)$ | | $(n, 128)$ |
| Dense | $(128, 256d)$ | $(n, 256d)$ |
| Reshape | | $(n, 32, 8d)$ |
| ReLU | | $(n, 32, 8d)$ |
| Trans Conv1D (Strde=4) | $(25, 8d, 4d)$ | $(n, 128, 4d)$ |
| ReLU | | $(n, 128, 4d)$ |
| Trans Conv1D (Strde=4) | $(25, 4d, 2d)$ | $(n, 512, 2d)$ |
| ReLU | | $(n, 512, 2d)$ |
| Trans Conv1D (Strde=4) | $(25, 2d, d)$ | $(n, 2048, d)$ |
| ReLU | | $(n, 2048, d)$ |
| Trans Conv1D (Strde=4) | $(25, d, d)$ | $(n, 8192, d)$ |
| Tanh | | $(n, 8192, d)$ |
| Conv1D (Stride=1) | $(256, 1, 1)$ | $(n, 8192, d)$ |

Table 4: Lightweight WaveGAN outputting 8 kHz sampling rate audio architecture. Scale d = 32.

| Operation | Kernel Size | Output Shape |
|---|---|---|
| Input audio | | $(n, l, 1)$ |
| Zero Insertion | | $(n, 2l, 1)$ |
| Conv1D (Stride = 1) | $(15, 1, 1)$ | $(n, 2l, 1)$ |
| Conv1D (Stirde = 4) | $(25, 1, d)$ | $(n, l2, d)$ |
| LReLU ($\alpha = 0.2$) | | $(n, l//2, d)$ |
| Conv1D (Stirde = 4) | $(25, d, 2d)$ | $(n, l//8, 2d)$ |
| LReLU ($\alpha = 0.2$) | | $(n, l//8, 2d)$ |
| Conv1D (Stirde = 4) | $(25, 2d, 4d)$ | $(n, l//32, 4d)$ |
| LReLU ($\alpha = 0.2$) | | $(n, l//32, 4d)$ |
| Noise Injection | $(4d, 4d + 1)$ | $(n, l//32, 4d + 1)$ |
| Trans Conv1D (Stride = 4) | $(25, 4d + 1, 2d)$ | $(n, l//8, 2d)$ |
| LReLU ($\alpha = 0.2$) | | $(n, l//8, 2d)$ |
| Concat | | $(n, l//8, 4d)$ |
| Trans Conv1D (Stride = 4) | $(25, 4d, d)$ | $(n, l//2, d)$ |
| LReLU ($\alpha = 0.2$) | | $(n, l//2, d)$ |
| Concat | | $(n, l//2, 2d)$ |
| Trans Conv1D (Stride = 4) | $(25, 2d, 1)$ | $(n, 2l, 1)$ |
| Tanh | | $(n, 2l, 1)$ |
| Conv1D (Stride=1) | $(m, 1, 1)$ | $(n, 8192, d)$ |

Table 5: Bandwidth extension module architecture.

## A.2 TRAINING DETAILS AND HYPERPARAMETERS

We considered other alternatives in the module architectures. If there were no skip-connections in the bandwidth extension module, the model could not generate valid waveform. Instance Normalization interrupted the training of the model. When we did not inject random noise on intermediate of the bandwidth extension module, the output quality was worse than the default architecture. Those models, that do not contain neural upsampling layer or were not trained progressively, fail to generate recognizable sounds.

| Name | Value |
|---|---|
| Channel size ($d$) (lightweight WaveGAN) | 64 |
| Channel size ($d$) (bandwidth extension module) | 16 |
| Batch size | 64 |
| Optimizer | Adam ($\alpha$ = 1e-4, $\beta1 = 0.0$ , $\beta1 = 0.9$) |
| Loss | WGAN-GP |
| WGAN-GP $\lambda$ | 10 |
| D updates per G module | 5 |

Table 6: Hyperparameters of PUGAN.

## A.3 ADDITIONAL RESULTS

To show our superior, we trained PUGAN on bird dataset , which has 44.1 kHz sampling rate audios (Vellinga & Planqué, 2015). We attached more BWEs to our model to generate 65,536 length of audio that means 1.48 sec in 44.1 kHz sampling rate. We downloaded by using the **R** library *warbleR* (Araya-Salas & Smith-Vidaurre, 2017), and we searched *singing* sounds of *robin* genus from we picked the instances which have 44.1 kHz sampling rate and quality A. The number of sounds is 23,920. See the demo site (https://pugan-iclr-demo.herokuapp.com/).

## A.4 ADDITIONAL FIGURES

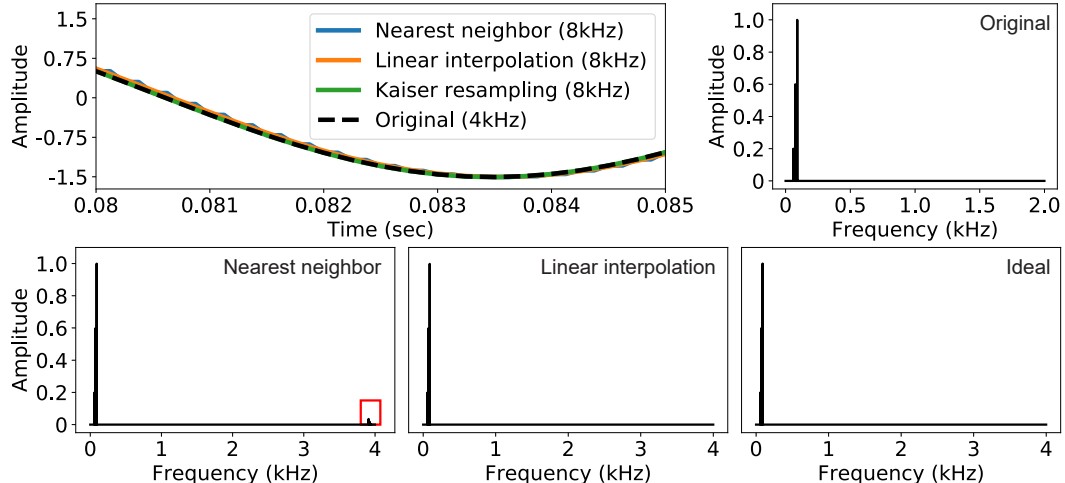

Figure 3: Original signal composed of three frequency components at 60, 75 and 90Hz - time domain plot (top left) and frequency domain (top right) plots. The plots below show the upsampling results using the nearest neighbor, linear interpolation and Kaiser resampling methods. The examples illustrate the occurrences of high frequency side bands (noise) similar to Fig. 1, however, the magnitude of noise decrease. It demonstrates that upsampling method is more important in high frequency than low one.

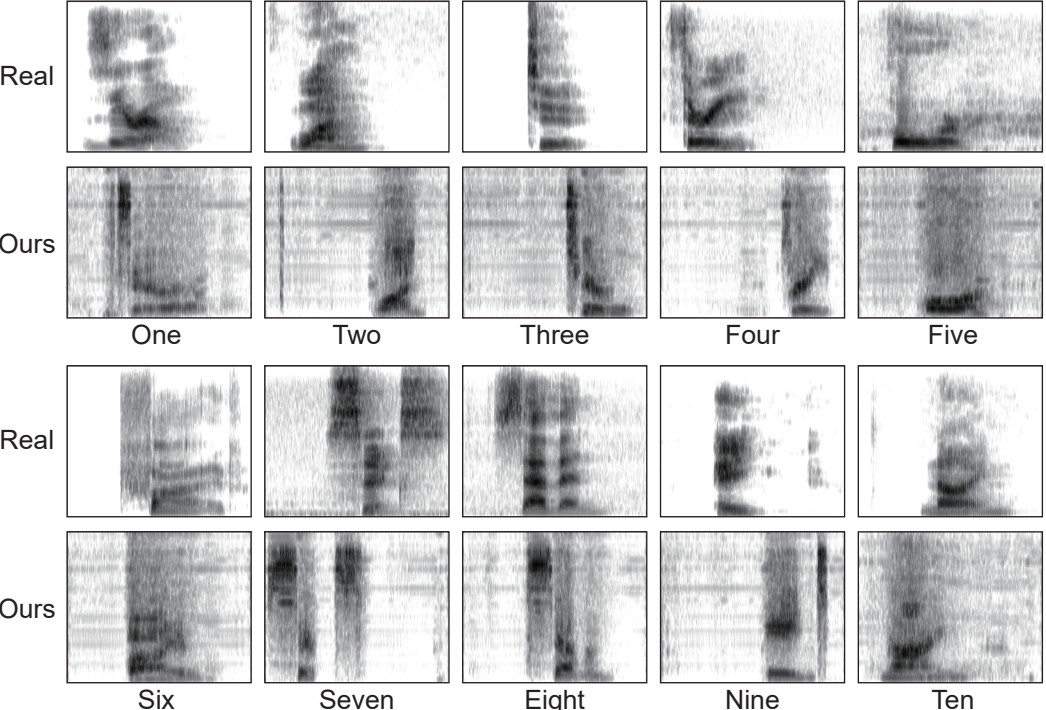

Figure 4: Samples from speech commands dataset and generated by PUGAN. Our model generate the same shape of formant for each voice data.

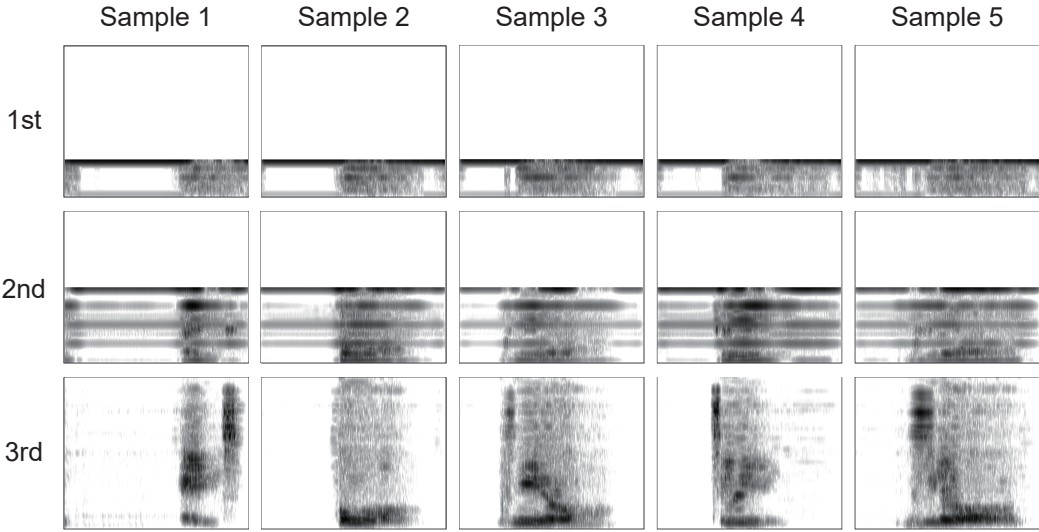

Figure 5: Each sample set shows the results generated by the generator for each step.

