# OpenReview forum: "Progressive Upsampling Audio Synthesis via Effective Adversarial Training"
_ICLR.cc/2020/Conference — Reject_

### Official Review · AnonReviewer2 · 2019-10-21
**Official Blind Review #2**

**Rating:** 1

**Review:**

Summary:

This work is a follow-up of WaveGAN. It uses the first few layers of the original WaveGAN to synthesize low resolution waveform (4kHz), and applies several bandwidth extension modules to progressively output the higher resolution raw audios.

pros:
- The proposed PUGAN has significantly smaller number of parameters than WaveGAN (e.g., 20x smaller).

cons:
- WaveGAN was a preliminary and encouraging trial for raw audio synthesis with GAN. Note that, its audio fidelity is far away from the state-of-the-art results and it was only tested on simple dataset (sounds of ten-digit commands). In contrast, the state-of-the-art autoregressive models (e.g., WaveNet) and parallel flow-based models (e.g., Parallel WaveNet) have been tested on challenging high-fidelity speech synthesis. As a result, one may focus on improving the audio fidelity of GAN on more challenging tasks. However, the proposed PUGAN was still tested on very simple dataset (sounds of ten-digit commands), and the quality of generated samples are only comparable to WaveGAN.

Detailed comment:

-- The attached samples are pretty noisy (e.g., noticeable artifacts on posted spectrograms). One may introduce the feature matching (e.g., STFT loss in ClariNet) as an auxiliary loss to improve the audio fidelity.

-- Did the authors try conditional generation, e.g., conditioned on the digit label? The posted failure cases and some samples tend to have overlapped sounds from different digits.

**Experience Assessment:**

I have published in this field for several years.

**Review Assessment: Checking Correctness Of Derivations And Theory:**

N/A

**Review Assessment: Checking Correctness Of Experiments:**

I assessed the sensibility of the experiments.

**Review Assessment: Thoroughness In Paper Reading:**

I read the paper at least twice and used my best judgement in assessing the paper.

---

> ### Author Response · Authors · 2019-11-15
> **response for the review**
>
> *** Note that, its audio fidelity is far away from the state-of-the-art results and it was only tested on simple dataset (sounds of ten-digit commands) ***
>
> As the reviewer mentioned, autoregressive models such as WaveNet performed well in the text-to-speech (TTS) applications. However, a comparative analysis of autoregressive models was done in the original WaveGAN paper, so it was not further evaluated in this paper. The current evaluations are based on a short sound of the duration of one second, where the previous study mentioned WaveNet did not perform very well, as can also be found in the official WaveGAN webpage at https://chrisdonahue.com/wavegan_examples/..
>
> While WaveNet is usually applied in TTS, the task of our scope is to learn to generate a short audio clip with no input text. That is, our goal is not to make sound close to WaveNet or long speech in a TTS setting. With this goal in mind, we also updated the demo webpage by training on the drum sound dataset and the greatest hits dataset. We have uploaded the generated samples to the demo page.
>
> TTS can be tackled by the autoregressive model, but our application is real-time sound effects generation in interactive content. Given user interaction in VR and AR, it creates the appropriate sound effect accordingly because GAN has a latent space. Also, as we mentioned in the introduction, it should generate at an interactive rate (about 10ms). There exist studies to speed up parallel wavenets, but the autoregressive model cannot solve this problem.
>
> ***The attached samples are pretty noisy (e.g., noticeable artifacts on posted spectrograms). One may introduce the feature matching (e.g., STFT loss in ClariNet) as an auxiliary loss to improve the audio fidelity. ***
>
> We use only one-dimensional waveforms for both the generator and discriminators. We think that the current goal of GANs to produce sound is to produce sound that is significant rather than reducing noise. The current accuracy is around 70%, and it is likely that a lot of research will be made to reduce the noise if the accuracy is somewhat higher. If we do not force the discriminator to receive a one-dimensional waveform, we may try to apply the STFT loss. However, looking at tSpecGAN and WaveGAN comparisons in WaveGAN paper and comparing accuracy in WaveGAN and Improved WaveGAN in this paper, noiselessness does not necessarily increase accuracy.
>
>
> *** Did the authors try conditional generation, e.g., conditioned on the digit label? The posted failure cases and some samples tend to have overlapped sounds from different digits. ***
>
> In this paper, we focused on the idea of progressive generation as the main idea, which shows that it helps to make it possible and better quality sound and reduce the number of parameters. Therefore, conditional generation is not considered in our problem setting.
>
> On the other hand, when we talk about overlapping voices of several digits or people, we have found that this problem occurs as we conduct experiments and write papers. However, this does not only occur in our PUGAN, but also in existing WaveGAN, and I would like to say that PUGAN does not have this problem as compared to WaveGAN.
>
> *
> Finally, we uploaded the code for our experiment to our demo page. One can check out the code from the link below and reproduce our experiment results by following the readme.txt file.
> https://pugan-iclr-demo.herokuapp.com/static/pugan-code.zip

---

### Official Review · AnonReviewer3 · 2019-10-24
**Official Blind Review #3**

**Rating:** 6

**Review:**

The authors detail PUGAN, architectural changes to models for raw waveform generation with GANs. They do a good job of motivating the challenge of raw audio generation with GANs and of methods for progressive training. PUGAN incorporates U-Net modules in the generator ("Bandwidth expansion"), sinc convolution as bandlimiting inputs to the generators, and the "style gan" type method of adding the noise at each level of the generator. Using listener studies and inception score, they show modest improvements over the state of the art (at time of submission), WaveGAN. Notably, their architecture is also more computation and parameter efficient.

The paper is well motivated and experiments are correct, but the quality improvements overall are a little underwhelming. To some extent, that can't be helped, and the authors wisely focus on the improvements in inference time as one of their central claims, and indeed produce evidence to support this claim.

More problematic, however, the paper motivates the problem of multi-scale generation of waveforms, but does not clearly show that the proposed architectures address those issues. The motivation in terms of interpolation artifacts and band-limited upsampling in Figure 1 give a misleading sense that Kaiser resampling is explicitly incorporated into the model. The authors argue that the U-Net layers implicitly learn an upsampling method, but the lack of model comparison / ablation makes it difficult to see if that really is the case. It would help support the claim to show samples from the model at different resolutions and demonstrate the lack of artifacts at each level. In the appendix, the authors mention that any alterations to the architecture resulted in failed training, but the lack of an ablation study makes it hard to know the relative value of each component. For example, it is unclear how important the sinc layers and bandlimiting are for the discriminators, the intermediate noise, and the use of the BWE architectures.

Despite these shortcomings I still recommend a weak accept, as the problem is difficult and the paper documents a well-motivated avenue for approaching it.


**Experience Assessment:**

I have published in this field for several years.

**Review Assessment: Checking Correctness Of Derivations And Theory:**

N/A

**Review Assessment: Checking Correctness Of Experiments:**

I carefully checked the experiments.

**Review Assessment: Thoroughness In Paper Reading:**

I read the paper thoroughly.

---

> ### Author Response · Authors · 2019-11-15
> **response for the reivew**
>
> *** More problematic, however, the paper motivates the problem of multi-scale generation of waveforms, but does not clearly show that the proposed architectures address those issues. ****
>
> To demonstrate the effectiveness of progressive training, we added a figure to the appendix, which includes a step-by-step result from the same latent vector. We added sound samples to the demo webpage. As a result, it can be seen that the high-frequency band is progressively learned. In the current PUGAN structure, we failed to train the model with no progressive training.
>
>
> *** The motivation in terms of interpolation artifacts and band-limited upsampling in Figure 1 give a misleading sense that Kaiser resampling is explicitly incorporated into the model.***
>
> We thank the reviewer for insightful suggestions. We renamed the kaiser resample as “ideal” in Figure 1 and changed the figure descriptions accordingly.
>
> *
> Finally, we uploaded the code for our experiment to our demo page. One can check out the code from the link below and reproduce our experiment results by following the readme.txt file.
> https://pugan-iclr-demo.herokuapp.com/static/pugan-code.zip

---

### Official Review · AnonReviewer1 · 2019-10-25
**Official Blind Review #1**

**Rating:** 3

**Review:**

The paper presents an approach based on generative adversarial models for the unconditional  generation of audio. The authors take inspiration from WaveGAN, to which they add more sophisticated upsampling blocks (called the bandwidth extension module) instead of transposed convolutions. They also propose to add a sinc convolution layer to the discriminators to improve training. Finally, they propose a progressive training scheme similar in spirit to the progressive training of GANs in images. Experiments are performed on generating audio pronunciation of digits, and the authors compare their work in terms of inception score, human evaluation and computation cost to WaveGAN.

As stated by the authors, one of the main motivation oof their approach is the reduction of computation cost. The motivation, expressed in terms of the 10ms "interactive rate" follows a good story. The measurements performed at the end of the paper show a ~ x2 performance gains compared to WaveGAN on CPU and about same running times on GPU, which is significant but not compelling.

Overall, I liked the story of the paper, but the paper lacks clarity and details. An important aspect of the paper is the progressive training, which is detailed nowhere (e.g., what is the stopping criterion to get to the next stage), should there be a special initialization of the last block of a new stage, etc.). The "Bandwidth extension module", which from my understanding is one of thee main contribution of the paper, is detailed in the appendix and comes essentially without justification. One of the main motivations of the paper is to be able to generate 44kHz audio, but the only results available at this resolution are inception scores that are below those of the 16kHz generation, which leaves open the question of whether the goal is effectively achieved.

I found the different versions of PUGAN difficult to read. The picture uses 2 blocks of bandwidth extension to generate 16kHz, whereas the evaluations are done with PUGAN-1 (1 block), which if I understand correctly is based on the lightweight WaveGAN that generates at 8kHz (whereas Section 4.1 suggests that 16kHz is generated from 4kHz generations by WaveGAN and two blocks). Also, the fact that evaluations are carried out with PUGAN-1 suggests that the progressive training does not really works well past a single block.

In the text it is suggested that spectral normalization and sinc conv on the discriminator is an "improvement" of WaveGAN, and an independent contribution of the article. While Table 2 clearly shows an improvement in terms of inception score, the human evaluation is not that clear: the accuracy of human labelers drops to 0.52 from 0.63 and the quality seems totally within the variance (2.7 +/- 1.5 vs 2.6 +/- 1.3). While Table 3 also shows clear improvement over the basic WaveGAN with the changes made by the authors (in particular in terms of win ratio vs PUGAN-1), the loss of accuracy should be discussed.

The performance obtained by PUGAN-1 is nonetheless noticeable -- +1 quality score over WaveGAN, and much better pairwise win ratios. Nonetheless, the paper lacks clarity and motivation for the exact form of the bandwidth extraction module, and does not fulfill its promises (importance of progressive training, high quality generation at 44kHz), so I am leaning towards rejection.



**Experience Assessment:**

I have published one or two papers in this area.

**Review Assessment: Checking Correctness Of Derivations And Theory:**

I assessed the sensibility of the derivations and theory.

**Review Assessment: Checking Correctness Of Experiments:**

I assessed the sensibility of the experiments.

**Review Assessment: Thoroughness In Paper Reading:**

I read the paper at least twice and used my best judgement in assessing the paper.

---

> ### Author Response · Authors · 2019-11-15
> **response for the review (1/2)**
>
> ***The measurements performed at the end of the paper show a ~ x2 performance gains compared to WaveGAN on CPU and about same running times on GPU, which is significant but not compelling.***
>
> While our model has a significantly smaller number of parameters, e.g., 10 times smaller than that of WaveGAN, but the inference time remained approximately the same in the previous version of the paper. However, after we newly optimize our implementation, we found 10 times of speed-up as shown in the following experimental results. In this experiment, we measured the computation time to generate 100 samples in a batch on GPU.  We revised the paper with new results.
>
> For the 16kHz audio,
> WaveGAN: 67.3ms
> PUGAN with one BWE: 5.2ms
> PUGAN with two BWEs: 5.8ms
> PUGAN with three BWEs: 4.9ms
>
> For the 44.1kHz audio,
> WaveGAN: 180.9ms
> PUGAN with three BWEs: 6.6ms
> PUGAN with four BWEs: 6.5ms
>
> Let us explain why the previous implementation had slow performance. We added “neural upsampling layer” to bandwidth extension module (BWE), which is composed of zero insertion and 1d convolutional layer (Section 4.2.2). When the module pads zero tensors, the model allocated new GPU memory and calculate again in the previous implementation. We simply changed the implementation to allocate zero tensors before the inference.
>
> Additionally, our model can be applied to interactive content media such as virtual reality and augmented reality that need a tremendous size of GPU memory for 3D rendering. PUGAN can significantly save the computation cost and the inference time for real-time use, as well as the memory usage.
>
> *** An important aspect of the paper is the progressive training, which is detailed nowhere (e.g., what is the stopping criterion to get to the next stage), should there be a special initialization of the last block of a new stage, etc.). ***
>
> We apologize for the confusion about progressive training procedure. We trained each step using the same number of epochs. We did not apply any special initialization technique.
>
> *** One of the main motivations of the paper is to be able to generate 44kHz audio, but the only results available at this resolution are inception scores that are below those of the 16kHz generation, which leaves open the question of whether the goal is effectively achieved. ***
>
> So far, we did not find a suitable high-resolution dataset for quantitative evaluation, which is why we did not evaluate it. Instead, we showed the successful results using the bird-singing dataset and posted the results in our demo webpage at https://pugan-iclr-demo.herokuapp.com/. As shown in the above table, for 44kHz audio, our model has 28 times of speed-up, compared to WaveGAN, e.g., 6.6ms vs. 180.9ms, while the number of parameters is 20 times smaller than WaveGAN.
>
>
> * PUGAN configurations *
>
> *** I found the different versions of PUGAN difficult to read. ***
>
> We thank the reviewer for pointing out the naming issue. The number that follows PUGAN indicated the number of BWE modules used. For improved readability, we changed the naming as “PUGAN with k-BWE”, e.g., PUGAN with one BWE and PUGAN with two BWEs
>
> *** Also, the fact that evaluations are carried out with PUGAN-1 suggests that the progressive training does not really works well past a single block. ***
>
> We conducted human evaluation with PUGAN with one BWE because it showed the best performance among PUGAN variations. PUGAN with two BWEs performed better than the original WaveGAN and the Improved WaveGAN. We updated the spectrograms of output audio at each step to demonstrate that the progressive training works.
>
> *** While Table 2 clearly shows an improvement in terms of inception score, the human evaluation is not that clear: the accuracy of human labelers drops to 0.52 from 0.63 and the quality seems totally within the variance (2.7 +/- 1.5 vs 2.6 +/- 1.3) ***
>
> *Accuracy of Improved WaveGAN accuracy
>
> We apologize for the confusion about accuracy and other measurements. We revised their explanation, as follows:
> Quality and accuracy were considered to measure different aspects. High accuracy but low quality means that the result sounds clear, but noisy or artificial. In the case of the Improved WaveGAN, there was no clear number, but that was the result of low noise and clean sound.
>
> On the other hand, we would like to emphasize that PUGAN variations have improved in all metrics.
>
> *** Nonetheless, the paper lacks clarity and motivation for the exact form of the bandwidth extraction module, and does not fulfill its promises (importance of progressive training, high quality generation at 44kHz), so I am leaning towards rejection. ***

---

> ### Author Response · Authors · 2019-11-15
> **response for the review (2/2)**
>
>
> To demonstrate the effectiveness of our progressive training, we added a figure to the appendix, which includes a step-by-step result from the same latent vector. We updated the demo webpage with these results. Overall, it can be seen that the high-frequency band is progressively learned. In the current PUGAN structure, we failed to train the model with no progressive training.
>
> *
> Finally, we uploaded the code for our experiment to our demo page. One can check out the code from the link below and reproduce our experiment results by following the readme.txt file.
> https://pugan-iclr-demo.herokuapp.com/static/pugan-code.zip

---

### Decision · Program_Chairs · 2019-12-19

**Decision:**

Reject

**Comment:**

Inspired by WaveGAN, this paper proposes a PUGAN to synthesizes high-quality audio in a raw waveform. The paper is well motivated. But all the reviewers find that the paper is lack of clarity and details, and there are some problems in the experiments.